# DYNAMIC STEERABLE FRAME NETWORKS

**Jörn-Henrik Jacobsen[1], Bert De Brabandere[2], Arnold W.M. Smeulders[1]**

[1]Department of Computer Science, University of Amsterdam
[2]ESAT-PSI, KU Leuven
{j.jacobsen,a.w.m.smeulders}@uva.nl
bert.debrabandere@esat.kuleuven.be

## ABSTRACT

Filters in a convolutional network are typically parametrized in a pixel basis. As an orthonormal basis, pixels may represent any arbitrary vector in $\mathbb{R}^n$. In this paper, we relax this orthonormality requirement and extend the set of viable bases to the generalized notion of frames. When applying suitable frame bases to ResNets on Cifar-10+ we demonstrate improved error rates by substitution only. By exploiting the transformation properties of such generalized bases, we arrive at steerable frames, that allow to continuously transform CNN filters under arbitrary Lie-groups. Further allowing us to locally separate pose from canonical appearance. We implement this in the Dynamic Steerable Frame Network, that dynamically estimates the transformations of filters, conditioned on its input. The derived method presents a hybrid of Dynamic Filter Networks and Spatial Transformer Networks that can be implemented in any convolutional architecture, as we illustrate in two examples. First, we illustrate estimation properties of steerable frames with a Dynamic Steerable Frame Network, compared to a Dynamic Filter Network on the task of edge detection, where we show clear advantages of the derived steerable frames. Lastly, we insert the Dynamic Steerable Frame Network as a module in a convolutional LSTM on the task of limited-data hand-gesture recognition from video and illustrate effective dynamic regularization and show clear advantages over Spatial Transformer Networks. In this paper, we have laid out the foundations of Frame-based convolutional networks and Dynamic Steerable Frame Networks while illustrating their advantages for continuously transforming features and data-efficient learning.

## 1 INTRODUCTION

For images, as well as any other sensory data, convolutional networks are typically learned from individual pixel values. Using them as a basis of the learned parameters is the standard approach for almost all CNNs. In this paper, we argue, that the pixel basis is not necessarily the best choice for representing signals. We show, that suitable alternatives yield increased classification performance by replacement only, while such a replacement adds additional properties to the learned filters that allow us to transform them under arbitrary pre-defined Lie groups.

From our perspective, the pixel values span an orthogonal basis for the filters in the network (in every layer). Such a pixel basis is complete as it may represent an arbitrary vector in $\mathbb{R}^n$ by linear combination, where $n$ is the dimensionality of the filter. In this paper we consider alternatives to this basis, both orthogonal bases, and non-orthogonal frames, arriving at superior expressiveness through steerable function spaces that allow us to transform filters locally and continuously, conditioned on their input.

Utilizing the steerability properties of frames in practice, we propose Dynamic Steerable Frame Networks (DSFNs) that fill the gap between Spatial Transformer Networks (STNs) (Jaderberg et al., 2015) and Dynamic Filter Networks (DFNs) (De Brabandere et al., 2016). STNs are not locally adaptive, thus they fail in many cases where it is not beneficial to transform the image globally as it would destroy discriminative information (multiple deformable objects, discriminative dynamic movements) or where global registration is performed as a preprocessing step (medical images).

DFNs are overcoming this restriction by locally transforming filters instead of globally transforming the whole feature stack as STNs do. However, DFNs are black boxes and not data-efficient, as they introduce many unconstrained parameters. Such a behavior is undesirable when data is limited and interpretability is key. DSFNs are locally adaptive, interpretable and data-efficient. They overcome the weaknesses of both approaches by combining their strengths, as illustrated in multiple experiments.

Our contributions:

- We argue that suitable frame bases are beneficial when representing sensory data compared to the commonly used pixel basis.
- Exploiting the transformation properties of frames further, we derive Dynamic Steerable Frame Networks that are able to continuously transform features locally and fill the gap between Spatial Transformer Networks and Dynamic Filter Networks.
- Dynamic Steerable Frame Networks learn to separate pose and feature. This enables the network to be locally equivariant or invariant with respect to certain feature poses, or even to perform in network quasi data-augmentation, while only the inputs and the backpropagated error signals determine which and to what extent these are applied.

We introduce the generalized notion of frames to CNNs that extend possible bases to learn from to non-orthogonal and overcomplete sets without loss in generalization. We show that many choices are possible, while overcomplete, non-orthogonal bases consistently outperform the pixel basis when applied to a ResNet (He et al., 2016) for image classification, as illustrated on Cifar-10+. We derive the Dynamic Steerable Frame Networks, based on the notion of steerable frames, that can locally adapt the filters in every feature map, conditioned on the input. We illustrate the strength of the approach in an edge detection task, where it outperforms a Dynamic Filter Network. We further show in a limited data video classification task, that Dynamic Steerable Frame Networks improve classification performance over Spatial Transformer Networks when global invariance is not desirable.

## 2 DYNAMICALLY STEERABLE FRAME NETWORKS

### 2.1 FRAMES

*Frames* are a natural generalization of orthogonal bases (Christensen, 2003). In frame terminology, an orthonormal basis is a Parseval-tight frame with unit norm. Every *tight frame* preserves the signal norm and exhibits perfect reconstruction. Frames can be seen as a superset of orthogonal bases in the sense that every basis is a frame, but not the reverse, see figure 1. The advantage of considering frames over orthogonal bases is that intrinsic signal properties can be spelled out explicitly in the new representation with the advantage, that these properties are directly accessible during learning. From an overcomplete representation, it will be more easily visible which part of the features is robust and which part is sensitive to accidental noise variations.

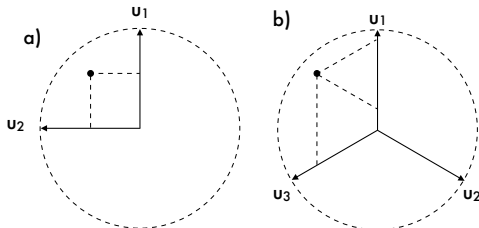

Figure 1: a) Is an orthonormal basis in $\mathbb{R}^2$, $u_1$ and $u_2$ are linearly independent and span the space of $\mathbb{R}^2$. A dot in this example represents a filter in a convolutional network with coefficients $\{v_1, v_2\}$. b) A tight frame in $\mathbb{R}^2$. $u_1$, $u_2$ and $u_3$ are linearly dependent. A dot in this example represents a convolutional filter with coefficients $\{v_1, v_2, v_3\}$. The frame is an overcomplete representation, again spanning $\mathbb{R}^2$ and again preserving the norm. Note that the set of filter coefficients as represented by the dot is not unique. Thus even if one $v$ is obstructed by noisy updates or measurements, the filter may still be robust.

In a standard convolutional network, a filter kernel is a linear combination over the standard basis for $l^2(\mathbb{N})$. The standard basis is composed from a delta function for every dimension and $W_i$ is the $i_{th}$ filter of the network with parameters $w_n^i$:

$$e_1 = \{1, 0, 0, ..., 0\}$$
$$e_2 = \{0, 1, 0, ..., 0\}$$
$$...$$
$$e_n = \{0, 0, 0, ..., 1\}$$
$$W_i = \sum_{n=1}^{N} w_n^i e_n$$

Without loss of generalization the orthonormal standard basis can be replaced by a frame to include non-orthogonality, overcompleteness, increased symmetries or steerability into the representation. Changing from the pixel to an arbitrary frame is as simple as replacing the pixel basis $e_n$ with a frame of choice with elements $v_n$ as follows:

$$W_i = \sum_{n=1}^{N} w_n^i v_n \tag{1}$$

where $w_1^i, ..., w_n^i$ are again the filter coefficients being learned.

In practice for CNNs working on images we investigate derived bases from steerability requirements, orthogonal polynomials, Framelets and members of the Gaussian derivative family. See figure 2 for a selection of frames

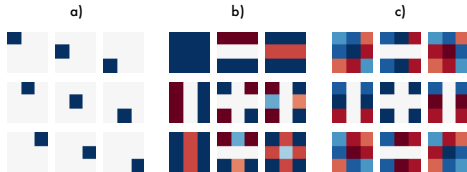

Figure 2: An illustrative plot of multiple 3x3 spanning sets: a) Pixel-basis, b) Orthogonal Polynomial, c) Non-orthogonal Frame. Note the increased symmetries in b) and c).

## 2.2 STEERING FRAMES UNDER ARBITRARY LIE-GROUPS

A pleasant property of many frames is steerability (Unser & Chenouard, 2013; Hel-Or & Teo, 1998; Michaelis & Sommer, 1995), the power of a function to represent transformed versions of itself by linear combination. The advantage of steerability in CNNs working on images is the ability to produce infinitely many transformed variants of a visual feature $f^\tau(x, y) \in \mathbb{R}^2 \to \mathbb{R}$ from its canonical appearance.

To achieve this goal we cast these variations as the result of the action of a family of transformations $g(\tau)$ on the canonical features $f(x, y)$, where $\tau \in R^k$ parametrizes these $k$-parameter transformations. If the problem at hand requires the distinction between multiple unknown poses of the same feature in a typical CNN they all have to be computed exhaustively to determine if a particular pose is present or not. Things go out of hand when the search space is a continuous transformation group, such as the Lie group of affine transformations, requiring $k \to \infty$ number of feature maps which is computationally intractable or requires expensive searches over all possible transformations (Gens & Domingos, 2014). One way out is to coarsely sample a few equally spaced points on the equivariant transformation manifold or to restrict the space to a smaller group (Cohen & Welling, 2016; Dieleman et al., 2016). What remains, however, is that the number of resulting feature maps for more general groups quickly becomes infeasible. An elegant way to overcome these limitations is the concept of steerability by (Freeman & Adelson, 1991; Perona, 1992; Unser & Chenouard, 2013) which is taken as inspiration here.

In this work, we focus on Lie groups. Transformations $g(\tau)$ over a range constitute a Lie group if they are closed under composition, they are associative, they are invertible, there exists an identity element, and their maps for inverse and composition are infinitely differentiable (Hel-Or & Teo, 1998). Teo and colleagues (Teo & Hel-Or, 1998) have given the following definition.

**Definition 1 (Steerability):** *A function $f(x,y)$: $\mathbb{R}^2 \to \mathbb{R}$ is steerable under a k-parameter Lie transformation group G if any transformation $g(\tau) \in G$ of f can be written as a linear combination of a fixed, finite set of frame functions $\phi_m(x,y)$:*

$$g(\tau)f(x,y) = \sum_{m=1}^{M} \beta_m(\tau)\phi_m(x,y) = \mathbf{B}^T(\tau)\Phi(x,y) \tag{2}$$

Where $\mathbf{B}^T(\tau)$ denote the collected steering functions describing the transformation and $\Phi(x,y)$ the collected steerable frame functions.

A function steerable under a k-parameter Lie group is capable of representing infinitely many states of a particular set of transformations. In many cases, only a finite set of frame functions is needed to represent these. In CNN terms, this means that a limited number of feature maps are sufficient to represent complete continuous transformation groups when the frame functions and the steering functions are chosen appropriately. Finding appropriate frame functions is the biggest challenge in steering arbitrary functions over arbitrary Lie groups.

To study the action of a Lie group $G$ on a function we use the close relation between the Lie group and its tangent space. The Algebra's tangent space spanned by the group's infinitesimal generators. The differential operators of the group action are obtained by computing the derivative of the group action with respect to its parameters at the identity element. A Lie Algebra can be considered as an "infinitesimal" Lie group. If the group is simply connected, the group action on a visual feature $f(x,y)$ can be obtained via the exponential map (Teo & Hel-Or, 1998):

$$g(\tau_1, ..., \tau_k)f(x,y) = e^{(\sum_{i=1}^{k} \tau_i L_i)} f(x,y) \tag{3}$$

where

$$e^{\tau_i L_i} = I + \tau_i L_i + \frac{1}{2!}\tau_i^2 L_i^2 + ... \tag{4}$$

where $L_i$ are the group's infinitesimal generators and $I$ is the identity element. This implies that one can compute the Taylor expansion with respect to the desired transformation group parameters to obtain elements of the group. If a finite frame set is equivariant towards the desired transformation group (it contains the orbit of the function to be steered), the series expansion yields linearly dependent elements after a finite number of steps. Then the frame is globally steerable under the desired transformation group. If this is not the case, as for example when scaling a Gaussian function, a finite frame set is only sufficient to accurately steer the function over a bounded interval, the function is locally steerable, but not globally.

## 2.3 SEPARATING POSE AND CANONICAL APPEARANCE

When training a CNN the functions represented by each feature naturally change from update to update. It is desirable to separate the frame functions from the effective features as learned by the network. In such a Structured Receptive Fields Network (RFNN) (Jacobsen et al., 2016), each filters parameters are not its mere pixel values, but the coefficients weighting the sum over a fixed frame set. Thus, analogous to equation 1, every effective filter $W_i(x,y)$ has the following form:

$$W_i(x,y) = w_1^i v_1 + w_2^i v_2 + ... + w_n^i v_n, \tag{5}$$

where $v_n$ denotes the $n_{th}$ element of the frame.

To be able to separate a features pose from its canonical appearance, we are interested in a steerable version of an arbitrary filter $W_i(x,y)$ under a k-parameter Lie group. From 5 follows:

$$g(\tau)W_i(x,y) = \sum_{n=1}^{N} w_n^i g(\tau)v_n^i. \tag{6}$$

And by substituting according to equation 2 it follows:

$$g(\tau)W_i(x,y) = \sum_{n=1}^{N} w_n^i \sum_{m=1}^{M} \beta_m(\tau)\phi_m(x,y). \tag{7}$$

Thus it is sufficient to determine the group action on the fixed frame by steering it to separate the canonical feature itself from its k-parameter variants, i.e. $v_n^i$ govern the weight of each frame coefficient to form a feature $W_i(x, y)$ and $\beta_m$ are the steering functions governing the transformation of $g(\tau)$ acting on $W_i(x, y)$ as a whole. From now on learning and transforming features amounts to a point-wise multiplication of frame coefficients with $\cos$, $\sin$ and $\exp$ activation functions, which is suitable for learning in a CNN.

## 2.4 DERIVING THE FRAME AND STEERING FUNCTIONS

Now the problem is reduced to finding a suitable frame as a function space underlying the learned filters. There are many approaches to derive a function space that is closed under the desired transformation group and as we show, many options give rise to bases that work considerably well when inserted into state-of-the-art CNNs. The most straightforward way is to derive it from the group's infinitesimal generators, for brevity we refer the interested reader to (Hel-Or & Teo, 1998) and directly cite some derived equivariant function spaces from the paper.

| Steerable Function Spaces | |
|---|---|
| x,y Translation | $x^p y^q e^{\alpha x + \beta y}$ |
| x,y Scaling | $x^\alpha y^\beta ln(x)^p ln(y)^q$ |
| Rotation & Uniform Scaling | $r^\alpha ln(r)^p e^{ik\theta}$ |
| x,y Translation & x,y Scaling & Rotation | $x^p y^q$ |

Table 1: Examples of function spaces closed under various non-Abelian multi-parameter groups, as derived in (Hel-Or & Teo, 1998). They can readily be used as a frame for CNNs by the procedure we derive here.

Once a frame is chosen, we can simply check if it is closed under the given transformation group by verifying for each generator $L_i$ that:

$$L_i \mathbf{\Phi}(x, y) = \mathbf{B}_i \mathbf{\Phi}(x, y), \tag{8}$$

where $\mathbf{B}_i$ is some finite dimensional $n \times n$ matrix. If this is the case, the function space is equivariant under the transformation group and we can compute the steering equations of the group composed of multiple generators as:

$$\mathbf{A}(\tau) = e^{\tau_k \mathbf{B}_k} \cdot ... \cdot e^{\tau_1 \mathbf{B}_1}. \tag{9}$$

To arrive at a practical solution, we have to consider the problem that locally bounded functions can not be steered globally with a finite steerable frame. To achieve a suitable approximation for our case, we separate scaling into two parts, an inner $\{\sigma_x, \sigma_y\}$ and an outer scale $\sigma_a$, where a stands for aperture. The inner scale can directly be steered via the above derivation and represents the slope of the local measurement taken by a filter, while the outer scale represents the size and shape of the filters receptive field. To achieve anisotropic receptive fields, we propose to first steer the scale at every pixel and steer the derived function space on this non-uniformly scaled grid, resulting in locally deformable receptive fields. Due to associativity of convolution, we can combine steering the derived function space and the receptive field scale into one operation. In this work, we use a second order approximation of the Gaussian that is capable of giving a good approximation to common CNN receptive field sizes 3x3, 5x5 and 7x7. For scaling over larger ranges, we recommend the spectral decomposition approach (Koutaki & Uchimura, 2014).

## 2.5 DYNAMIC STEERABLE FRAME NETWORKS

Estimating the local pose of a feature from a steerable function space is analytically intractable in the case of most multi-parameter groups. In this paper, we introduce the Dynamic Steerable Frame Network that combines the advantages of steerable function spaces with the power of neural network function estimators, by estimating pose parameters from a function space equivariant under the transformation group at hand. Specifically, our architecture is inspired by the recently introduced Dynamic Filter Networks (De Brabandere et al., 2016). The Dynamic Filter Network (DFN) generates one feature per location in a feature map, which boils down to a locally connected convolution layer, for which the parameters are generated by a different network that estimates them from the input, yielding a different filter kernel for every location in the input.

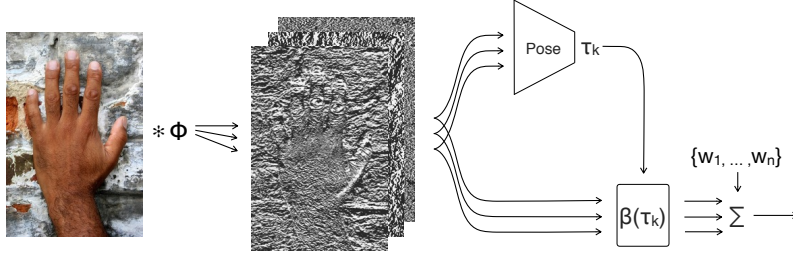

Figure 3: The Dynamic Steerable Frame Network. The network transforms an input image to a steerable frame $\mathbf{\Phi}$ (here an example with 3 frame functions) and estimates the local feature pose at each location in this equivariant space with a small pose estimating network. Then it outputs a set of pose coordinates $\tau_{\mathbf{k}}$, that are dependent on the group parametrization chosen. They are inserted into the matrix of steering equations $\beta(\tau)$ and applied to the frame $\mathbf{\Phi}$, yielding the locally steered frame. In the same operation, we integrate the weights $w_n$, that govern the feature maps canonical feature appearance, these are the weights learned by a normal CNN. The Dynamic Steerable Frame Network can decide to commute with a set of poses, to be invariant to them, to only look for certain poses or to act like a normal CNN, where each feature map has one pose and one canonical appearance assigned to itself. This is only determined by the input data and the backpropagated error signals.

The DFN takes the form:
$$O(x, y) = F_\tau^{x,y}(I(x, y)), \qquad (10)$$
where $F_\tau^{x,y}$ are generated by another network from the input. We propose the Dynamic Steerable Frame Network, where the parameters $\theta$ that condition the filter are pose transformation parameters of the steerable function space, estimated from the input, similar to how it is done in the Spatial Transformer Networks, just that in our case we aim for locally adaptive filters. The filters $F_\tau^{x,y}$ share the same set of weights in the whole feature map, so they represent the same canonical appearance everywhere. While their local pose is dynamically estimated by a Pose-Generating Network $\Psi$ that takes the form:
$$\tau(x, y) = \Psi(I(x, y)). \qquad (11)$$
Thus, the canonical appearance is translation invariant, but its geometrical pose is not. In terms of equation 5, this means the set of $w_n^i$ is fixed, but the frame $v_n^{\tau(x,y)}$ is locally transformed under a pre-defined k-parameter group with parameters $\tau$. See figure 5 for an illustration.

The method consists of two parts: i) A Pose-Generating network estimating local pose parameters of a feature conditioned on the input from a steerable input space. ii) A Dynamic Filtering mechanism, convolving transformed versions of a feature with every location in the input feature map, based on the estimates of the pose generating network. Due to linearity of convolution, we can first perform a transformation of the input into the steerable frame space and in this space we perform i) and ii) as point-wise multiplications.

# 3 RELATED WORK

Steerable Filters is a concept established early for signal processing. Initially introduced by (Freeman & Adelson, 1991), the concept was extended to the Steerable Pyramid by (Simoncelli & Freeman, 1995) and further extended to a Lie-group formulation by (Hel-Or & Teo, 1998; Michaelis & Sommer, 1995). Further, steerability has recently been extended to tight frames, presenting Simoncelli's Steerable Pyramid and multiple other Wavelets arising as a special case of the non-orthogonal Riesz transform (Unser & Chenouard, 2013). Steerable pyramids have been applied to CNNs as a pre-processing step (Xue et al., 2016), but have not yet been learnable. We incorporate steerable frames in CNNs to increase their de facto expressiveness and to allow them to learn their configurations, rather than picking them a priori.

Convolutional Networks with alternative bases have been proposed with various degrees of flexibility. A number of works utilizes change of basis to stabilize training and increase convergence behavior (Rippel et al., 2015; Arjovsky et al., 2015). Another line of research is concerned with complex-valued CNNs, either learned (Tygert et al., 2016), or fully designed like the Scattering networks (Bruna & Mallat, 2013; Oyallon & Mallat, 2015).

Scattering, as well as the complex-valued networks, rest upon a direct connection between the signal processing literature and CNNs. Inspired by the former, Structured Receptive Field Networks are learned from an overcomplete multi-scale frame, effectively improving performance for small datasets due to restricted feature spaces (Jacobsen et al., 2016). Also related is the work on Group-equivariant CNNs (Cohen & Welling, 2016) and Cyclic Pooling (Dieleman et al., 2016), where equivariance towards the dihedral group is theoretically guaranteed, yielding increased accuracy. Inspired by CNNs learned from alternative bases, we introduce the general principle of Frame-based convolutional networks that allow for non-orthogonal, overcomplete and steerable feature spaces.

Another way to impose structure onto CNN representations and subsequently increase their data-efficiency is to incorporate explicit geometrical transformations into them. Either by learning transformation operators and group representations (Cohen et al., 2014; Wang et al., 2009). Or by pre-defining the possible transformations, as done in Transforming Autoencoders (Hinton et al., 2011), which map their inputs from the image to pose space through a neural network. The Spatial Transformer Networks (Jaderberg et al., 2015) learn global transformation parameters in a similar way while applying them to a nonlinear co-registration of the feature stack to some learned pose. This yields especially high performance on tasks where centering the objects is beneficial. Dynamic Filter Networks move one step further and estimate filters for each location, conditioned on their input. These approaches are all dynamic in a sense that they condition their parameters on the input appearance. We combine the idea of Dynamic Filter Networks with explicit pose prediction into Dynamic Steerable Frame Networks that can estimate poses from continuous input space, conditioned on the input. As such, we overcome the difficulty of estimating local pose, while being able to separate pose and feature learning globally.

## 4 EXPERIMENTS

### 4.1 GENERALIZING PIXELS TO FRAMES ON CIFAR-10+

To show the validity of general frame representations, we compare different bases in a state-of-the-art pre-activation deep residual network architecture (He et al., 2016) on the Cifar-10+ (Krizhevsky & Hinton, 2009) dataset with moderate data augmentation of crops and flips.

| Error on Cifar10+ | | | |
|---|---|---|---|
| Method | Pixel | Image Frame | Naive Frame |
| ResNet-20 | 7.85% | 7.61% | 8.97% |
| ResNet-56 | 6.68% | 6.08% | 7.30% |
| ResNet-110 | 5.84% | 5.34% | 6.96% |
| Densenet K12 L40 | 5.28% | 4.99% | 6.39% |
| Densenet K12 L100 | 4.16% | 3.78% | 5.21% |

Table 2: Results on Cifar10 with moderate data-augmentation (crops/flips) with the recently introduced pre-activation Residual network and Densenet with the standard pixel-basis, a steerable frame basis designed for natural images and the naive steerable $x^p y^q$ frame from table 3 that does not take natural image statistics into account. The natural image statistics based frame outperforms the pixel-basis consistently, while the naive frame consinstently performs about 1% worse than the baseline, highlighting the benefit of a frame suitable for the type of input data.

We evaluated our approach on multiple networks and network sizes. The setup used for the ResNet is as described in (He et al., 2016). The batch size is chosen to be 64 and we train for 164 epochs with the described learning rate decrease. The ResNet architectures used are without bottlenecks having 20, 56 and 110 layers. For the Densenets we follow (Huang et al., 2016) and evaluate on the K=12 and L=40, and the K=12 and L=100 models. We run our experiments in Keras (Chollet, 2015) and Tensorflow (Abadi et al., 2016). In the first experiment, we run the models on the standard pixel basis to get a viable baseline. Secondly, we replace the pixel-basis with widely-used frames that take natural image statistics into account, namely non-orthogonal, overcomplete Gaussian derivatives (Florack et al., 1992) and non-orthogonal framelets (Daubechies et al., 2003) in an alternating fashion, yielding superior performance compared to the pixel-basis by replacement only.

We also show that the naive $x^p y^q$ frame (see table 1) performs consistently worse than the other two choices, as it does not take natural image properties into account, while it is important to mention that this 1% performance decrease also comes with additional properties that might be highly beneficial in particular tasks. We have also found orthogonal polynomials to not work very well (around 3% performance decrease), which is in line with our expectation that suitable frames should take natural image statistics into account. 2D frames are generated from 1D functions via the following generating process:

$$Frame = \{v_0, v_1, v_2, v_3\} \otimes \{v_0, v_1, v_2, v_3\}^T. \tag{12}$$

The results are reported in table 2. The fact that the pixel-basis can be replaced by steerable frames and performance even improves when the frame is chosen well, is remarkable, as this means every filter in the CNN enjoys additional properties, while performance improves in the standard setting already and finding suitable frames is not more expensive than running the same smallest CNN as many times as one has frames to choose from, as the performance we observed was consistent across multiple model sizes. Frame-based CNNs run at the same runtime as vanilla CNNs.

## 4.2 DYNAMIC STEERABLE FRAME NETWORKS

In this section we report two experiments. The first experiment is an edge detection task, highlighting the difference between our approach and multiple baselines in a fine-grained pixel-wise labeling task. In the second experiment, we apply a 2D convolutional LSTM on a small hand gesture recognition video dataset to illustrate how the Dynamic Steerable Frame Network regularizes the model effectively and to illustrate its benefits over Spatial Transform Networks.

The model used in both experiments is learned from a steerable Gauss-Hermite frame. The Dynamic Steerable Frame Network consists of three processing steps. 1) Change to frame space on the input, 2) the Pose-Generating network estimates the pose from this transformed input, outputting a set of pose variables for each location in the image. 3) the steering functions derived in section 2.4 are applied to these pose variable maps and effectively act as nonlinear pose-parametrized activation functions that regularize the Pose-Generating network to output an explicitly interpretable pose space. Finally, a 1x1 convolution layer is applied to the already transformed output maps, representing the weights $w_n^i$, governing the canonical appearance of the $i_{th}$ feature map, see also figure 5. Dynamic Steerable Frame Networks run at the same computational cost as vanilla Dynamic Filter Networks.

### 4.2.1 EDGE DETECTION

In this experiment, we compare a Dynamic Filter Network (De Brabandere et al., 2016) baseline with an autoencoder and a Dynamic Steerable Frame Network on the task of edge detection. The problem is formulated as a pixel-wise classification task and reported is the root mean-squared error on an unseen test set. The labels are the edges. The dataset is infinite, as we produce random blobs and create the edge labels with a standard scikit image function. The standard DFN can freely learn an input layer with 2 filters and 3 subsequent 1x1 layers that can non-linearly recombine the inputs, whereas the Frame DFN receives a steerable frame as an input, allowing it to leverage the fine-grained orientation information without the need to learn it. The Dynamic Steerable Frame Network has the exact same architecture as the DFN but is geometrically regularized on its output as can be seen in figure 5, as an input it receives a first order Gauss-Hermite frame that can be steered globally towards rotation and locally towards scale.

Location varying methods are clearly superior in this task, compared to the location invariant autoencoder. The DFN increases its performance substantially when getting the steerable frame as an input, indicating its inability to learn a continuously transforming frame by itself. Finally, the Dynamic Steerable Frame Network clearly outperforms all baselines due to its ability to continuously transform its filters in a well-regularized manner. As an extra, we get the local feature pose for free from the output of the DSFN, the baseline has no notion of an explicit pose parameter, see figure 4.

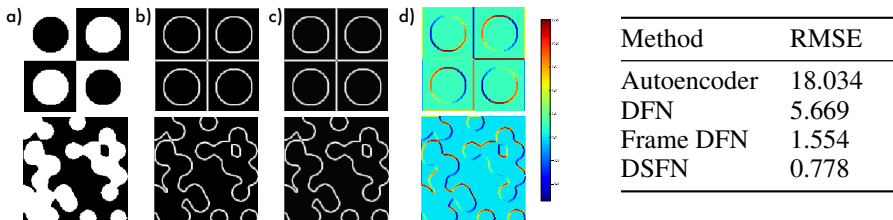

| Method | RMSE |
|---|---|
| Autoencoder | 18.034 |
| DFN | 5.669 |
| Frame DFN | 1.554 |
| DSFN | 0.778 |

Figure 4: Results on the edge detection task. Top is an illustration of a test image, bottom one sample from the actual infinite dataset, reported is root mean squared error. Autoencoder denotes a vanilla location invariant autoencoder. DFN denotes the plain Dynamic Filter Network, Frame DFN denotes a DFN whos input is a frame, DSFN denotes the Dynamic Steerable Frame Network. a) is the input, b) is the label, c) the prediction and d) the angular pose variable. d) is an output we get for free when training DSFNs, while a DFN has no notion of interpretable angle variables. Location varying methods clearly outperform the static autoencoder, while learning the DFN from a steerable frame increases performance again substantially. The DSFN substantially outperforms all other methods due to its continuously transforming input and output space.

### 4.2.2 SMALL SCALE VIDEO CLASSIFICATION

To show the ability of the Dynamic Steerable Frame Network to effectively regularize in dynamic settings where poses play an important role and where Spatial Transformer Networks do not work well, we apply it on the task of Hand-Gesture Recognition. Namely, on the Cambridge Hand-Gesture dataset (Kim & Cipolla, 2009), consisting of 9 classes of hand movement and poses in 900 videos, we use 750 for training, 50 for validation and 100 for testing. The dataset is very small and contains classes where global movement plays an important role and thus provides a good test bed to show effectiveness of the DSFN regularization ability compared to Spatial Transformers.

| convLSTM | 1 Layer | 2 Layer | rot/scale-DSFN | rot/scale-STN | affine-STN |
|---|---|---|---|---|---|
| # Params | 905k | 913k | 907k | 971k | 1037k |
| Accuracy | 35.42% | 39.31% | 62.18% | 21.34% | 12.21 % |

Table 3: Results on the Cambridge Hand-Gesture Recognition dataset, to illustrate the effectiveness of pose regularization provided by the Dynamic Steerable Frame Network. Adding the DSFN module to the convLSTM drastically improves performance. Increasing the capacity of the baseline to two layers, does not make up for the difference in performance, while adding the STN to the convLSTM decreases performance significantly, as the STN does not manage to learn meaningful global transformations that do not remove the class-specific information content. This is further substantiated by an increased performance when removing the ability to shear and translate the input from the STN. The DSFN outperforms all other approaches while only adding 2k free parameters to the baseline.

Our baseline model is a convolutional LSTM with 10 output feature maps, batch normalization and a dense layer for classification. As a second baseline, we increase the capacity of the model by adding a second LSTM layer and a second batch normalization step. We combine two instances of a Spatial Transformer Network with a convolutional LSTM, one that can perform full affine transformations and one that is restricted to rotation and scaling. The DSFN module is applied to the input layer of the smaller model with 4 output feature maps. The setup of the steerable frame used in this model is a Gauss-Hermite frame.

The steerability is uniquely parametrized as: $\{B_s B_\theta\}$. Allowing for scaling and rotation. Both Spatial Transformer Networks do not manage to learn useful warps of the input image and therefore decrease performance of the baseline. The affine model only manages to correctly classify multiple instances of a static class that has no movement information related to its label, while the rot/scale model increases performance, but still does not manage to learn useful scalings or rotations. The DSFN manages to learn locally rotation and scale invariant filters, that follow the boundaries and other features across the video as desired. Results are reported in Table 3.

For visualizations of the learned transformations, see Appendix A. The results illustrate the effectiveness of the DSFN to regularize the LSTM on a small-scale task where mostly local invariance is desired, but global invariance destroys most of the class-specific information. The DSFN improves performance over the baseline by about 22%, while the Spatial Transformer Network decreases performance by about 15%, or even to random in the full-affine case.

## 5 DISCUSSION

We have introduced the notion of Frame-based convolutional networks. Our experiments illustrate that a simple replacement of the standard basis by a frame suitable for natural images leads to increased performance.

The insight that multiple frames can be considered as viable spanning sets for CNN representations leads us to steerable frames whose properties we exploit explicitly in our derived Dynamic Steerable Frame Networks, such that they can readily be accessed during training. The proposed method is a hybrid of Dynamic Filter Networks and Spatial Transformer Networks, enabling locally adaptive filtering with geometrical constraints.

We illustrate the effectiveness of the approach on an edge detection task, that requires fine-grained pixel-wise labeling, where Dynamic Steerable Frame Networks outperform a standard Dynamic Filter Network and an autoencoder baseline. Further, we illustrate the ability of the Dynamic Steerable Frame Network to regularize recurrent networks in a small-data video classification scenario where Spatial Transformer Networks fail to learn meaningful transformations.

Future work is to apply the model to other problem domains like egocentric video, robotics applications, as well as volumetric medical imaging videos of moving organs. We expect our Dynamic Steerable Frame Network approach to be beneficial in any problem where spatiotemporal continuity, data-efficiency, or interpretable pose spaces are key.

## ACKNOWLEDGEMENTS

We would like to thank Edouard Oyallon and Taco Cohen for insightful comments and discussions.

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

## APPENDIX A    VISUALIZING DSFN AND STN TRANSFORMATIONS

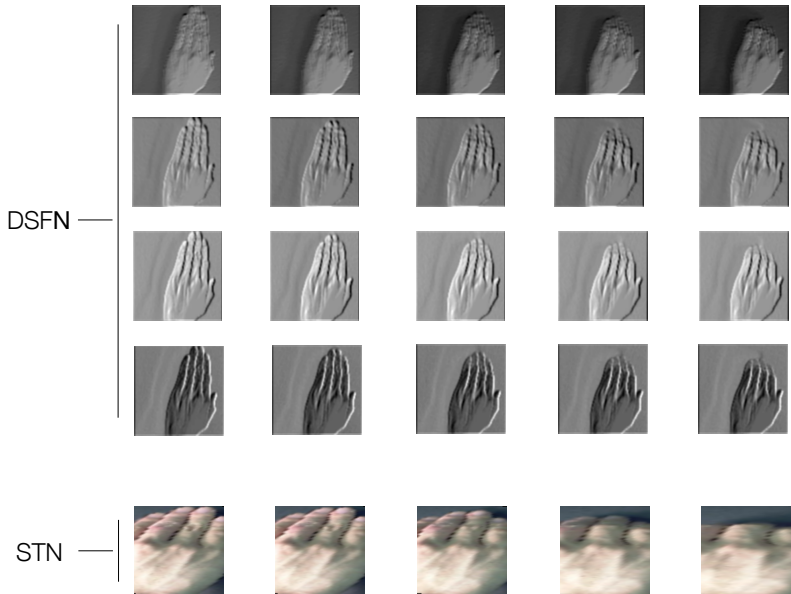

Figure 5: Visualizations of the learned transformations by the Dynamic Steerable Frame Network (DSFN top) and the Spatial Transformer Network (STN bottom) on the hand-gesture recognition dataset. The STN zooms and rotates the hands arbitrarily and apparently removes important information content thereby, leading to low classification accuracy. The DSFN acts locally and adaptively filters the hands in multiple ways. Note that the DSFN did not learn fully rotation invariant filters in all 4 cases, but in 3 cases produces different filter responses for different sides of the hand. However, it does follow the contours of the hand and segments the borders from the background. This indicates that full rotation invariance is not suitable for this task. This would be hard to assess if one had to choose the degree of invariance a priori, while the DSFN has the means to learn the necessary amount of local invariance.

## APPENDIX B    EQUIVARIANCE PROOF & STEERING EQUATION DERIVATION

To prove that a frame is equivariant with respect to the action of a group transformation, determined by its generator $L_i$, we simply have to show that:

$$L_i \mathbf{\Phi}(x, y) = \mathbf{B}_i \mathbf{\Phi}(x, y), \tag{13}$$

Where $\mathbf{B}_i$ is some $n \times n$ matrix.

In case of the Hermite polynomials (here considered up to second order), we have:

$$\mathbf{\Phi}(x, y) = \{1, x, y, x^2 - 1, xy, y^2 - 1\}. \tag{14}$$

To verify that these functions span an equivariant function space with respect to rotation, we apply the generator of rotations to the frame and verify that equation 13 holds. The generator of rotations

in the plane is given by $L_r = -x\frac{d}{dy} + y\frac{d}{dx}$, applied to each frame element, we get:

$$L_r \mathbf{\Phi}(x,y) = \begin{bmatrix} 0 \\ y \\ -x \\ 2xy \\ -x^2 + y^2 \\ -2xy \end{bmatrix} = \mathbf{B}_r \begin{bmatrix} 1 \\ x \\ y \\ x^2 - 1 \\ xy \\ y^2 - 1 \end{bmatrix}. \tag{15}$$

It is straightforward to solve this linear system and obtain the $6 \times 6$ matrix $B_r$:

$$\mathbf{B}_r = \begin{bmatrix} 0 & 0 & 0 & 0 & 0 & 0 \\ 0 & 0 & 1 & 0 & 0 & 0 \\ 0 & -1 & 0 & 0 & 0 & 0 \\ 0 & 0 & 0 & 0 & 2 & 0 \\ 0 & 0 & 0 & -1 & 0 & 1 \\ 0 & 0 & 0 & 0 & -2 & 0 \end{bmatrix}. \quad \square$$

Thus, we have proven that the function space is closed under the action of the group and the Hermite polynomials constitute an equivariant function space with respect to rotation.

Subsequently, the exponential map directly yields the steering equations collected in the interpolation matrix $\mathbf{A}^\theta$, that can rotate the whole frame by $\theta$:

$$\mathbf{\Phi}^\theta(x,y) = e^{\theta \mathbf{B}_r} \mathbf{\Phi}(x,y) = \mathbf{A}^\theta \mathbf{\Phi}(x,y),$$

$$\mathbf{\Phi}^\theta(x,y) = \begin{bmatrix} 1 & 0 & 0 & 0 & 0 & 0 \\ 0 & \cos\theta & \sin\theta & 0 & 0 & 0 \\ 0 & -\sin\theta & \cos\theta & 0 & 0 & 0 \\ 0 & 0 & 0 & \frac{1}{2} + \frac{1}{2}\cos 2\theta & \sin 2\theta & \frac{1}{2} - \frac{1}{2}\cos 2\theta \\ 0 & 0 & 0 & -\frac{1}{2}\sin 2\theta & \cos 2\theta & \frac{1}{2}\sin 2\theta \\ 0 & 0 & 0 & \frac{1}{2} - \frac{1}{2}\cos 2\theta & -\sin 2\theta & \frac{1}{2} + \frac{1}{2}\cos 2\theta \end{bmatrix} \begin{bmatrix} 1 \\ x \\ y \\ x^2 - 1 \\ xy \\ y^2 - 1 \end{bmatrix}.$$

$\mathbf{\Phi}^\theta(x,y)$ is the frame rotated by some angle $\theta$. Combining this result with equation 7, gives us the possibility to rotate any learned feature by arbitrary and continuous angles $\theta$. The whole procedure is completely analogous for any other Lie group transformation. Further, k-parameter transformation groups can be composed according to equation 9 from smaller groups. Here an example of the general linear group of rotation, anisotropic scalings and skew:

$$\mathbf{\Phi}^{\{\theta_1, s_x, s_y, \theta_2\}}(x,y) = \mathbf{A}^{\{\theta_1, s_x, s_y, \theta_2\}} \mathbf{\Phi}(x,y) = e^{\theta_2 \mathbf{B}_r} \cdot e^{s_x \mathbf{B}_{s_x}} \cdot e^{s_y \mathbf{B}_{s_y}} \cdot e^{\theta_1 \mathbf{B}_r} \mathbf{\Phi}(x,y). \tag{16}$$

## APPENDIX C    BACKPROPAGATION THROUGH STEERABLE FILTERS

Will be added to final manuscript.

