# Peer review of "Dynamic Steerable Frame Networks"

_ICLR 2017 — rejected_

[Official Review · AnonReviewer2 · rating 7 · confidence 3 · 16 Dec 2016]
**Interesting approach for adaptable convolutional filters**

This works applies steerable frames for various tasks where convolutional neural networks with location invariant operators are traditionally applied. Authors provide a detailed overview of steerable frames followed with an experimental section which applies dynamic steerable network to small machine learning problems where the steerability is conceptually useful.

Even though the evaluation is performed only on few small tasks, the reason why more tasks were not evaluated is that piece-wise pose invariance is needed only for a subset of tasks. The fact, that simply using overcomplete bases as a sort of "feature pre-processing" improves the results for already highly optimized ResNet and DenseNet architectures is quite interesting achievement.

For the edge detection, a relatively hard baseline is selected - the Dynamic Filter Networks, which already attempts to achieve position invariant filters. The fact that DSFN improves the performance on this task verifies that regressing the parametrization of the steerable filters yields better results than regressing the filters directly.

In the last experiment authors apply the network to video classification using LSTMs and they show that the improved performance is not due to increased capacity of the network.

In general, it is quite interesting work. Even though it does not offer ground-breaking results (mainly in a sense of not performing experiments on larger tasks), it is theoretically interesting and shows promising results.

There are few minor issues and suggestions related to the paper:
* For the LSTM experiment, in order to be more exact, it would be useful to include information about total number of parameters, as the network which estimates the pose also increases the number of parameters.
* Would it be possible to provide more details about how the back-propagation is done through the steerable filters?
* For the Edge Detection experiment, it would be useful to provide results for some standard baseline - e.g. CNN with a similar number of parameters. Simply to see how useful it is to have location-variant filters for this task.
* The last sentence in second paragraph on page 1 is missing a verb. Also it is maybe unnecessary.
* The hyphenation for ConvNet is incorrect on multiple places (probably `\hyphenation{Conv-Net}` would fix it).

[Official Review · AnonReviewer3 · rating 4 · confidence 3 · 20 Dec 2016]
**No Title**

I sincerely apologize for the late review!

The first part has a strong emphasis on the technical part. It could benefit from some high level arguments on what the method aims to achieve, what limitation is there to overcome. I may have misunderstood the contribution (in which case please correct me) that the main novel part of the paper is the suggestion to learn the group parameterizations instead of pre-fixing them. So instead of applying it to common spatial filters as in De Brabandere et al., it is applied to Steerable Frames?

The first contribution suggests that "general frame bases are better suited to represent sensory input data than the commonly used pixel basis.". The experiments on Cifar10+ indicate that this is not true in general. Considering the basis as a hyper-parameter, expensive search has to be conducted to find that the Gauss-Frame gives better results. I assume this does not suggest that the Gauss-Frame is always better, at least there is weak evidence on a single network presented. Maybe the first contribution has to be re-stated. Further is the "Pixel" network representation corrected for the larger number of parameters. As someone who is interested in using this, what are the runtime considerations? 

I would strongly suggest to improve Fig.3. The Figure uses "w" several times in different notations and depictions. It mixes boxes, single symbols and illustrative figures. It took some time to decipher the Figure and its flow. 


Summary: The paper is sufficiently clear, technical at many places and readability can be improved. E.g., the introduction of frames in the beginning lacks motivation and is rather unclear to someone new to this concept. The work falls in the general category of methods that impose knowledge about filter transformations into the network architecture. For me that has always two sides, the algorithmic and technical part (there are several ways to do this) and the practical side (should I do it)? This is a possible approach to this problem but after the paper I was a bit wondering what I have learned, I am certainly not inspired based on the content of the paper to integrate or build on this work. I am lacking insights into transformational parameters that are relevant for a problem. While the spatial transformer network paper was weaker on the technical elegance side, it provided exactly this: an insight into the feature transformation learned by the algorithm. I am missing this here, e.g., from Table 2  I learn that among four choices one works empirically better. What is destroyed by the x^py^p and Hermite frames that the ResNet is *not* able to recover from? You can construct network architectures that are the superset of both, so that inferior performance could be avoided. 

The algorithm is clear but it is similar to the Dynamic Filter Networks paper. And I am unfortunately not convinced about the usefulness of this particular formulation. I'd expect a stronger paper with more insights into transformations and comparisons to standard techniques, a clear delineation of when this is advised.

[Official Review · AnonReviewer1 · rating 5 · confidence 4 · 26 Dec 2016]

This paper presents an improved formulation of CNN, aiming to separate geometric transformation from inherent features. The network can estimate the transformation of filters given the input images. 

This work is based on a solid technical foundation and is motivated by a plausible rationale. Yet, the value of this work in practice is subject to questions:

(1) It relies on the assumption that the input image is subject to a transformation on a certain Lie group (locally). Do such transformations constitute real challenges in practice? State-of-the-art CNNs, e.g. ResNet, are already quite resilient to such local deformations. What such components would add to the state of the art? Limited experiments on Cifar-10 does not seem to provide a very strong argument.

(2) The computational cost is not discussed.

[Public Comment · Joern-Henrik Jacobsen · 19 Jan 2017 (modified: 20 Jan 2017)]
**Revision Summary**

Dear Reviewers,

Thank you once again for your helpful comments, suggestions and time.

We have revised the manuscript based on your reviews.

1. We have reworked the text according to your suggestions. We revised the whole introduction to put more focus on high-level arguments for our proposed method and to get rid of unnecessary technicality. We have rewritten the experimental section to focus more on comparison with standard methods. And we have updated other parts, including the abstract accordingly.

2. We have added multiple experiments to highlight the benefit of our method and to show a wider comparison to other methods.

- We have added additional experiments with SOTA Densenets to the Cifar10+ table, with additional results on the "naive frame", where the "image frame" outperforms the Densenet baseline as well as it did outperform the ResNets, highlighting that our observations generalize across multiple architectures.

- We have added a Spatial Transformer Network to the hand-gesture recognition task in two settings, a full-affine STN, and a restricted scale-rotation STN. In both cases, the STN fails to learn meaningful global warps that benefit the classification. However, restricting the full-affine to scale-rotation increases performance by 10%, indicating that less global invariance is better in a task where most class-specific information is encoded in global appearance and movement and only local invariance is desirable. Our proposed DSFN increases performance above all baselines by at least 22%.

- We have added parameter counts for the models evaluated on the hand-gesture recognition task, showing that the DSFN comes at a very small parameter cost.

- To facilitate model insight, we have added visualizations of the transformations learned by the Spatial Transformer Networks and the Dynamic Steerable Frame Networks to the appendix. Showing, that the DSFN learns local invariance, while the STN zooms and rotates the videos in what seems to be arbitrary ways.

- On the edge-detection task, we have added an experiment where we provide a frame as an input to a Dynamic Filter Network, to assess if the DFN benefits from a steerable function space as input. The DFNs performance increases substantially if we do so, indicating, that the DFN was not able to learn a continuously transforming frame on its own. Our proposed DSFN again cuts the error of the strongest DFN result by half, illustrating that contiuously transforming input and output spaces are superior when precise local adaption is needed.

- As requested by reviewer 2, we have also added an additional Autoencoder baseline to the results on the edge-detection task. As expected, it illustrates that location invariant filtering performs poorly on such a task.

3. We have updated figure 3 as suggested by reviewer 3, removed redundancy and moved it as close as possible to the actual computational flow of the proposed method.

We would like to thank the reviewers once again for their insightful reviews and hope that we were able to answer all their questions.

Let us know if you have any further questions or remarks.

[Final Decision · Program Chairs · 06 Feb 2017]
**ICLR committee final decision**

This paper studies how to incorporate local invariance to geometric transformations into a CNN pipeline. It proposes steerable filter banks as the ground-bed to measure and produce such local invariance, building on previous work from the same authors as well as the Spatial Transformer Networks. Preliminary experiments on several tasks requiring different levels of local invariance are presented. 
 
 The reviewers had varying opinions about this work; all acknowledged the potential benefits of the approach, while some of them raised questions about the significance and usefulness of the approach. The authors were very responsive during the rebuttal phase and took into account all the feedback. 
 
 Based on the technical content of the paper and the reviewers opinion, the AC recommends rejection. Since this decision is not consensual among all reviewers, please let me explain it in more detail.
 
 - The current manuscript does not provide a clear description of the new model in the context of the related works it builds upon (the so-called dynamic filter networks and the spatial transformer networks). The paper spends almost 4 pages with a technical exposition on steerable frames, covering basic material from signal processing. While this might indeed be a good introduction to readers not familiar with the concept of steerable filters, the fact is that it obfuscates the real contributions of the paper. which are not clearly stated. In fact, the model is presented between equations (10) and (11), but it is not clear from these equations what specifically differentiates the dsfn from the other two models -- the reader has to do some digging in order to uncover the differences (which are important). 
 
 Besides this clarity issue, the paper does not offer any insight as to how the 'Pose generating network' Psi is supposed to estimate the pose parameters. Which architecture? what is the underlying estimation problem it is trying to solve, and why do we expect this problem to be efficiently estimated with a neural network? when are the pose parameters uniquely determined? how does this network deal with the aperture effects (i.e. the situations where there is no unicity in determining a specific pose) ?
 Currently, the reader has no access to these questions, which are to some extent at the core of the proposed technique.